# Sparse Hypergraph Community Detection Thresholds in Stochastic Block Model

**Erchuan Zhang**
School of Science
Edith Cowan University
erchuan.zhang@ecu.edu.au

**David Suter**
School of Science
Edith Cowan University
d.suter@ecu.edu.au

**Giang Truong**
School of Science
Edith Cowan University
h.truong@ecu.edu.au

**Syed Zulqarnain Gilani**
School of Science
Edith Cowan University
s.gilani@ecu.edu.au

## Abstract

Community detection in random graphs or hypergraphs is an interesting fundamental problem in statistics, machine learning and computer vision. When the hypergraphs are generated by a *stochastic block model*, the existence of a sharp threshold on the model parameters for community detection was conjectured by Angelini et al. 2015. In this paper, we confirm the positive part of the conjecture, the possibility of non-trivial reconstruction above the threshold, for the case of two blocks. We do so by comparing the hypergraph stochastic block model with its Erdös-Rényi counterpart. We also obtain estimates for the parameters of the hypergraph stochastic block model. The methods developed in this paper are generalised from the study of sparse random graphs by Mossel et al. 2015 and are motivated by the work of Yuan et al. 2022. Furthermore, we present some discussion on the negative part of the conjecture, i.e., non-reconstruction of community structures.

## 1 Introduction

Community detection, or clustering, aims to identify groups sharing similar properties from a global population. It is a fundamentally important problem in statistics, network analysis and computer vision [15, 34, 11, 28, 26]. Many clustering algorithms have been developed based on graphs in which each edge contains exactly two vertices representing pairwise relationships on data. However, real-world data are often much more complicated than the ordinary graph structures. For example, in the co-authorship network [9, 24, 27], the number of co-authors may vary from paper to paper so that one can hardly study edges consisting of two co-authors only.

To represent higher order relationships among data, *hypergraphs* have been proposed to model data and have been shown to have some advantages over graphs [36]. In the co-authorship example above, a generalised edge (called *hyperedge*) allows us to consider the connectivity of arbitrarily many co-authors. In recent years, hypergraphs have been widely used to model complex data relationships, for instance, in robust multi-structure clustering [26], bioinformatics [31], and social networks extraction [35]. However, from both theoretical and methodological sides, the complexity of hypergraphs creates much more challenges in the sense that many analysis tools developed in the graph case can not be easily generalised to the hypergraph case. In general, analysing the spectral properties of the adjacency tensor of a hypergraph is more difficult than studying the spectrum of the adjacency matrix of a graph.

36th Conference on Neural Information Processing Systems (NeurIPS 2022).

The *stochastic block model* (SBM), also known as the *planted partition model*, was initially proposed by Holland et al. [17] to model random graphs with community structures. The general idea behind this model is to label each vertex as a community member identically and independently, and then construct a random graph based on the membership and some given probabilities. It becomes the well-known Erdös-Rényi graph model if we ignore the membership restriction. The recovery and/or detection problem, based on the SBM, is to find the community membership when the number of vertices is sufficiently large. SBM often serves as a benchmark for clustering algorithms on graph data. Likewise, an analogous model called the *hypergraph stochastic block model* (HSBM) was proposed to model higher-order relationships for random hypergraphs [13]. In what follows, we will present a brief introduction to hypergraphs, HSBM and its associated detection problem.

## 1.1 The stochastic block model for hypergraphs

A *hypergraph* is a pair $H = (\mathcal{V}, \mathcal{E})$, where $\mathcal{V}$ is a set of vertices and $\mathcal{E}$ is a set of *hyperedges*, which is a collection of subsets of $\mathcal{V}$. The *degree* of a vertex $v \in \mathcal{V}$ is the number of hyperedges in $H$ that contains $v$. The *degree* of a hyperedge $e \in \mathcal{E}$ is the number of vertices contained in $e$. The hypergraph $H$ is called *d-uniform* if the degree of every hyperedge $e \in \mathcal{E}$ is $d$. A 2-uniform hypergraph is a graph. An *l-cycle* in $H$ is a cyclic ordering of a subset of the vertex set, and of hyperedges, where any two adjacent hyperedges have exactly $l$ common vertices. We call an $l$-cycle *loose* if $l = 1$. For instance, let $H_3 = (\mathcal{V}, \mathcal{E})$ be a 3-uniform hypergraph, where $\mathcal{V} = \{v_1, v_2, \cdots, v_7\}$, $\mathcal{E} = \{(v_i, v_j, v_k) | 1 \leq i < j < k \leq 7\}$. Then $\{(v_1, v_2, v_3), (v_3, v_4, v_5), (v_5, v_6, v_1)\}$ is a loose cycle.

There are many ways to represent a hypergraph. For instance, the *incidence matrix* of a hypergraph $H$ is defined as $A \in \{0, 1\}^{|\mathcal{V}| \times |\mathcal{E}|}$, where $A(v, e) = 1$ if the vertex $v \in \mathcal{V}$ is contained in the hyperedge $e \in \mathcal{E}$, and 0 otherwise. The *adjacency tensor* of $H$ is defined as $T$, where $T_{i_1, \cdots, i_d} = 1$ if $\{i_1, \cdots, i_d\}$ is a hyperedge of $H$, and 0 otherwise. The *adjacency matrix* of $H$ is defined as $B = (B_{ij})_{|\mathcal{V}| \times |\mathcal{V}|}$, where $B_{ij} = \sum_{e:\{i,j\} \in e} T_e$ if $i \neq j$, and 0 otherwise, here $e$ is a hyperedge and $T_e$ is an element in its associated adjacency tensor. Unfortunately, most of the computations involving the adjacency tensor are NP-hard, for examples, determining the rank-1 approximation of a tensor, whether a tensor has a given spectral norm [16].

For simplicity, from now on, we specify our $d$-uniform hypergraph stochastic block model with two roughly balanced communities. Considering non-uniform hypergraphs or hypergraphs with multi-communities may require extra effort, which will be reserved for future work. Let $H = ([n], \mathcal{E})$ be a $d$-uniform hypergraph with vertex set $[n] := \{1, 2, \cdots, n\}$ and hyperedge set $\mathcal{E}$, $\sigma := (\sigma_1, \cdots, \sigma_n) = \{+1, -1\}^n$ the spins on $[n]$, which means each vertex $i \in [n]$ is assigned with a spin $\sigma_i \in \{+1, -1\}$. Let $\mathcal{S}_n$ be the set of all pairs $(H, \sigma)$, we can generate a random pair $(H, \sigma)$ from the finite set $\mathcal{S}_n$ as follows ([13, 14, 19, 25]):

- First generate i.i.d random variables $\sigma_i \in \{+1, -1\}$ uniformly for each $i \in [n]$.

- Then, for the obtained $\sigma = (\sigma_1, \cdots, \sigma_n)$, we generate a random $d$-uniform hypergraph $H$ where an hyperedge $e = \{i_1, \cdots, i_d\}$ is included independently with probability $p_n$ if $\sigma_{i_1} = \cdots = \sigma_{i_d}$, and with probability $q_n$ otherwise, where $0 < q_n < p_n < 1$ ($p_n, q_n$ possibly depending on $n$).

We denote by $\mathcal{H}_d(n, p_n, q_n)$ a $d$-uniform hypergraph generated by the process above. (With a little notation abuse, we sometimes denote the distribution that such hypergraphs follow by the same notation). In particular, $\mathcal{H}_2(n, p_n, q_n)$ is a random graph generated by the graph version of the SBM.

Suppose $\mathcal{C}_1 = \{i \in [n] | \sigma_i = +1\}$ and $\mathcal{C}_2 = \{i \in [n] | \sigma_i = -1\}$ are two communities in the hypergraph $H$. The goal of community detection is to estimate the unknown spin $\sigma$ up to a sign flip by observing $H$ only from a sample $(H, \sigma)$ drawn from $\mathcal{H}_d(n, p_n, q_n)$, i.e., to find estimators $\hat{\sigma}_i \in \{+1, -1\}$ that are correlated with the true partition $\sigma_i$. By correlation, we mean there exists a constant $c \in (0, 1)$, for any $\epsilon > 0$,

$$\lim_{n \to \infty} \mathbb{P}(\{|\mathcal{O}_n(\hat{\sigma}, \sigma) - c| > \epsilon\} \cap \{|\mathcal{O}_n(\hat{\sigma}, \sigma) + c| > \epsilon\}) = 0,$$

where $\mathcal{O}_n(\hat{\sigma}, \sigma) := \frac{1}{n} \sum_{i \in [n]} \sigma_i \hat{\sigma}_i$ is the empirical overlap between $\sigma$ and $\hat{\sigma}$.

Recall the definitions of exact recovery and almost exact recovery of communities in the literature (for example, see [25]). By *exact recovery*, we mean finding a spin vector $\hat{\sigma} \in \{+1, -1\}^n$ such that

$$\lim_{n \to \infty} \mathbb{P}(\{\mathcal{O}_n(\hat{\sigma}, \sigma) = 1\} \cup \{\mathcal{O}_n(\hat{\sigma}, \sigma) = -1\}) = 1.$$

That is, the estimated spin vector $\hat{\sigma}$ is either exactly the true spin vector $\sigma$ or exactly the negative true spin vector $-\sigma$. By *almost exact recovery*, we mean finding a spin vector $\hat{\sigma} \in \{+1, -1\}^n$ such that, for any $\epsilon > 0$,

$$\lim_{n \to \infty} \mathbb{P}(\{|\mathcal{O}_n(\hat{\sigma}, \sigma) - 1| > \epsilon\} \cap \{|\mathcal{O}_n(\hat{\sigma}, \sigma) + 1| > \epsilon\}) = 0.$$

That is, the empirical overlap $\mathcal{O}_n(\hat{\sigma}, \sigma)$ is in $[-1, -1 + \epsilon) \cup (1 - \epsilon, 1]$ asymptotically almost surely. This case almost exactly recovers the community, as indicated by the terminology, compared with the definition of exact recovery.

The study of *sparse* graphs/hypergraphs as well as those with constant average degree, is well motivated from the viewpoint of real networks. For instance, many large social networks like Twitter or LinkedIn, may have several millions of nodes but only ten times more edges [20]. A coauthorship network may have millions of nodes, but the average degree is no more than 20 in general [24]. Therefore, it is natural to consider the sparse HSBM with $p_n$ and $q_n$ of the order $\mathcal{O}\left(1/n^{d-1}\right)$.

Decelle et al. [7] conjectured the existence of a sharp threshold, called *Kesten-Stigun threshold*, of detection in the sparse graph SBM based on some ideas from statistical physics. More precisely, if $p_n = \frac{a}{n}$, $q_n = \frac{b}{n}$, where $a, b$ are constants independent of $n$, then detection is possible if and only if $(a - b)^2 > 2(a + b)$. This conjecture was confirmed in [21, 22, 23, 6], for examples, by studying the spectra of the non-backtracking [6], self-avoiding [21] or graph powering [1] matrices. Moreover, some efficient algorithms were proposed to achieve detection at the threshold [6, 29, 1].

In the case of the hypergraph SBM, Angelini et al. [2] conjectured the existence of a sharp threshold on the model parameters, for community detection in the sparse hypergraph. If $p_n = \frac{a}{\binom{n}{d-1}}$, $q_n = \frac{b}{\binom{n}{d-1}}$, i.e., the hypergraph $H \sim \mathcal{H}_d(n, p_n, q_n)$ has constant expected vertex degree. Pal et al. [25] proved the positive part of the conjecture, the detection is possible if $\beta^2 > \alpha$, where $\alpha = (d-1)\frac{a + (2^{d-1} - 1)b}{2^{d-1}}$, $\beta = (d-1)\frac{a-b}{2^{d-1}}$. Their method is based on the spectral analysis of the so-called *self-avoiding matrix* associated with the hypergraph, which is a generalisation to random hypergraphs of the method developed by Massoulié [21] for sparse random graphs. Moreover, the consistent estimators for model parameters $a$ and $b$ are still unknown[1].

In the present paper, we reconfirm the positive part of the conjecture by comparing the sparse HSBM with an appropriate Erdös-Rényi hypergraph model (see Theorem 1.1), which has also been used by Yuan et al. [34]. Further, we give consistent estimators for $a$ and $b$ if the detection is possible. Our methods are motivated by the study of the community detection problem on the graph SBM conducted by Mossel et al. [22]. The main results of the present paper are summarised as follows.

## 1.2 Main results

Let $\mathcal{H}_d\left(n, \frac{p_n + (2^{d-1} - 1)q_n}{2^{d-1}}\right)$ be the Erdös-Rényi model in which each hyperedge is included with a common probability $\frac{p_n + (2^{d-1} - 1)q_n}{2^{d-1}}$, which is chosen to make sure this model has the same average degree as $\mathcal{H}_d(n, p_n, q_n)$. In other words, $\mathcal{H}_d\left(n, \frac{p_n + (2^{d-1} - 1)q_n}{2^{d-1}}\right)$ represents a $d$-uniform random hypergraph without any community structures. In particular, $\mathcal{H}_2\left(n, \frac{p_n + q_n}{2}\right)$ is the traditional Erdös-Rényi random graph that has been extensively studied in the graph literature [4, 12, 8].

Let $\mathbb{P}_n$ and $\tilde{\mathbb{P}}_n$ denote the probability measures with respect to $\mathcal{H}_d(n, p_n, q_n)$ and $\mathcal{H}_d\left(n, \frac{p_n + (2^{d-1} - 1)q_n}{2^{d-1}}\right)$, respectively. Then $\mathbb{P}_n$ and $\tilde{\mathbb{P}}_n$ are said to be *mutually contiguous* if for any measurable sets $A_n$, $\mathbb{P}_n(A_n) \to 0$ if and only if $\tilde{\mathbb{P}}_n(A_n) \to 0$ as $n \to \infty$. $\mathbb{P}_n$ and $\tilde{\mathbb{P}}_n$ are said to be

---

[1] By some personal correspondence with Ludovic Stephan and Yizhe Zhu, we now realized that the algorithm provided in [25] implied some way to find a way to estimate the parameters $a, b$, i.e., Theorem 6.1 of [25] gave an estimate of $\alpha$ and a lower bound on $\beta$. However, it is not efficient and precise as the method in [30], where the two leading eigenvalues of the non-backtracking operator give an estimation of $a$ and $b$.

*asymptotically orthogonal* if there exists a sequence of measurable sets $A_n$ such that $\mathbb{P}_n(A_n) \to 0$ and $\tilde{\mathbb{P}}_n(A_n) \to 1$ as $n \to \infty$. From now on, we fix the following notation,

$$p_n := \frac{a}{\binom{n}{d-1}}, \quad q_n := \frac{b}{\binom{n}{d-1}}, \tag{1}$$

where $a > b > 0$ are some constants independent of $n$, $d \geq 2$ is an integer. Define

$$\alpha := (d-1)\frac{a + (2^{d-1}-1)b}{2^{d-1}}, \quad \beta := (d-1)\frac{a-b}{2^{d-1}}, \tag{2}$$

where $\alpha$ measures the expected degree of any vertex and $\beta$ measures the discrepancy between the number of neighbors with different signs of any vertex.

**Theorem 1.1.** *If $\beta^2 > \alpha$, then $\mathbb{P}_n$ and $\tilde{\mathbb{P}}_n$ are asymptotically orthogonal.*

**Theorem 1.2.** *Let $X_{\zeta_n}$ be the number of loose cycles of length $\zeta_n$ and define*

$$\hat{\alpha}_n := \frac{d|\mathcal{E}|}{\binom{n}{d-1}}, \quad \hat{\beta}_n := (2\zeta_n X_{\zeta_n} - \hat{\alpha}_n^{\zeta_n})^{\frac{1}{\zeta_n}},$$

*where $\zeta_n = \lfloor \log^{1/4} n \rfloor$, $|\mathcal{E}|$ is the number of observed hyperedges, then $\hat{a}_n = \frac{1}{d-1}\left(\hat{\alpha}_n + (2^{d-1}-1)\hat{\beta}_n\right)$ and $\hat{b}_n = \frac{1}{d-1}\left(\hat{\alpha}_n - \hat{\beta}_n\right)$ are consistent estimators for $a$ and $b$, respectively. Namely, $\hat{a}_n \to a$ and $\hat{b}_n \to b$ in probability as $n \to \infty$.*

When $d = 2$, the condition $\beta^2 = \alpha$ becomes $(a-b)^2 = 2(a+b)$, which is the threshold for community detection in the (graph-based) SBM. Readers may refer to [21, 22, 23] for more detail. Intuitively speaking, Theorem 1.1 says that $\mathcal{H}_d(n, p_n, q_n)$ and $\mathcal{H}_d\left(n, \frac{p_n + (2^{d-1}-1)q_n}{2^{d-1}}\right)$ are asymptotically almost surely distinguishable above the threshold (i.e., can distinguish a community with structure from a random hypergraph with the same average degree). Note that Theorem 1.1 also appeared in [34] though there are some subtleties. In [34], the setting is $p_n = \frac{a}{n^{d-1}}$, $q_n = \frac{b}{n^{d-1}}$, which is the same order as (1) (only differs by a constant factor $(d-1)!$). Moreover, $\kappa$ in Equation (4) in Section 2.2 in [34] corresponds to our $\beta^2/\alpha$ here. We generally follow the proof techniques of [34]. The difference is that they show convergence to a Normal instead of a Poisson distribution for the model $\mathcal{H}_d(n, p_n, q_n)$. If we take $k = 2$, $m = 3$ in Theorem 2.6 in [33], which means the 2-block, 3-uniform HSBM, they showed a sharp phase transition: when $\kappa < 1$, $\mathcal{H}_d(n, p_n, q_n)$ and $\mathcal{H}_d\left(n, \frac{p_n + (2^{d-1}-1)q_n}{2^{d-1}}\right)$ are indistinguishable and distinguishable when $\kappa > 1$.

There are two types of problems or results: (a) find a way to distinguish HSBM with a corresponding Erdös-Rényi hypergraph model and find a way to estimate the model parameters $a, b$; (b) develop an algorithm to output a partition $\hat{\sigma}$ that is correlated (at least $1/2 + \epsilon$ portion of the labels are correct up to a sign flip) to the ground truth labeling $\sigma$. Our work addresses (a), which does not imply any way to find an estimator of the partition for (b). We get an estimator for the model parameters $a, b$ by counting cycles, but it does not provide a way to find a partition $\hat{\sigma}$. Also, solving (b) does not immediately imply solving (a), although most algorithms for (b) (such as the ones in [25, 30]) may imply some way to solve (a).

The remainder of the present paper is organised as follows: In section 2, we study the probability distribution of the number of $\zeta_n$-loose cycle $X_{\zeta_n}$ and then prove our main results based on the techniques developed by Mossel et al. [22]. In section 3, we show some building blocks of proving our main results, which followed the proof techniques of [34]. In section 4, we discuss the barriers on showing the negative side of the threshold conjecture, i.e., non-reconstruction of community structures. Section 5 concludes the paper, including some further discussion.

## 2 The distribution of loose cycles

In this section, we mainly show that the number of $\zeta_n$-loose cycles of $H \sim \mathcal{H}_d(n, p_n, q_n)$ or $H \sim \mathcal{H}_d\left(n, \frac{p_n + (2^{d-1}-1)q_n}{2^{d-1}}\right)$ is asymptotically approximately Poisson-distributed. In general inhomogeneous graphs, Bollobás et al. [5] demonstrated that the number of cycles asymptotically

follows the Poisson distribution. We borrow their proof techniques to get a similar conclusion in the hypergraph case. Then, by comparing the first two moments of the number of $\zeta_n$-loose cycles, we can find that $\mathcal{H}_d(n, p_n, q_n)$ and $\mathcal{H}_d\left(n, \frac{p_n + (2^{d-1}-1)q_n}{2^{d-1}}\right)$ are asymptotically orthogonal if $\beta^2 > \alpha$.

**Theorem 2.1.** *Let $X_{\zeta_n}$ be the number of $\zeta_n$-loose cycle of a hypergraph $H$. Suppose $\zeta_n = \mathcal{O}\left(\log^{1/4}(n)\right)$.*

*(1) If $H \sim \mathcal{H}_d(n, p_n, q_n)$, then $X_{\zeta_n} \xrightarrow{d} Pois\left(\frac{\alpha^{\zeta_n} + \beta^{\zeta_n}}{2\zeta_n}\right)$;*

*(2) If $H \sim \mathcal{H}_d\left(n, \frac{p_n + (2^{d-1}-1)q_n}{2^{d-1}}\right)$, then $X_{\zeta_n} \xrightarrow{d} Pois\left(\frac{\alpha^{\zeta_n}}{2\zeta_n}\right)$,*

*where $\xrightarrow{d}$ denotes convergent in distribution.*

For any non-negative integer $s$, let $[X]_s$ be the product $X(X-1)\cdots(X-s+1)$.

**Lemma 2.2** (Lemma 2.8 in [32]). *Let $\lambda_1, \cdots, \lambda_k$ be some set of fixed non-negative reals, and let $X_{1,n}, \cdots, X_{k,n}$ be non-negative integer random variables defined on the same space $\mathcal{G}_n$ for each $n$. If for each fixed set of non-negative integers $r_1, \cdots, r_k$,*

$$\lim_{n\to\infty} \mathbb{E}\left(\prod_{i=1}^{k}[X_{i,n}]_{r_i}\right) \to \prod_{i=1}^{k}\lambda_i^{r_i},$$

*then the variables $X_{1,n}, \cdots, X_{k,n}$ are asymptotically independent Poisson with means $\lambda_i$.*

By Lemma 2.2, if we can prove the expectation $\mathbb{E}[X_{\zeta_n}]_r \to \mu^r$, then $X_{\zeta_n} \xrightarrow{d} \text{Pois}(\mu)$, which will be shown in the next section.

With the preparation provided by establishing Theorem 2.1, now we can prove our first main result, Theorem 1.1. We use $\mathbb{P}_n(\cdot)$, $\mathbb{E}(\cdot)$, $\text{Var}(\cdot)$ to denote probability, mean and variation with respect to $\mathcal{H}_d(n, p_n, q_n)$ while notations with tilde above denote their counterparts with respect to $\mathcal{H}_d\left(n, \frac{p_n + (2^{d-1}-1)q_n}{2^{d-1}}\right)$.

*Proof of Theorem 1.1.* Since $\beta^2 > \alpha$, there exists a constant $\rho$ satisfying $\beta > \rho > \sqrt{\alpha}$. Then $\mathbb{E}X_{\zeta_n} = \text{Var}X_{\zeta_n} = o(\rho^{2\zeta_n})$ and $\tilde{\mathbb{E}}X_{\zeta_n} = \tilde{\text{Var}}X_{\zeta_n} = o(\rho^{2\zeta_n})$ as $n \to \infty$.

Let $A_n := \{X_{\zeta_n} \leq \tilde{\mathbb{E}}X_{\zeta_n} + \rho^{\zeta_n}\}$, by Chebyshev's inequality,

$$\tilde{\mathbb{P}}_n(A_n) = \tilde{\mathbb{P}}_n\left(\frac{X_{\zeta_n} - \tilde{\mathbb{E}}X_{\zeta_n}}{\sqrt{\tilde{\text{Var}}X_{\zeta_n}}} \leq \frac{\rho^{\zeta_n}}{\sqrt{\tilde{\text{Var}}X_{\zeta_n}}}\right) \geq 1 - \frac{\tilde{\text{Var}}X_{\zeta_n}}{\rho^{2\zeta_n}} \to 1, \quad \text{as } n \to \infty. \quad (3)$$

Since $\mathbb{E}X_{\zeta_n} - \tilde{\mathbb{E}}X_{\zeta_n} \sim \frac{\beta^{\zeta_n}}{2\zeta_n} = \omega(\rho^{\zeta_n})$, then for larger enough $\zeta_n$, $\mathbb{E}X_{\zeta_n} - \rho^{\zeta_n} \geq \tilde{\mathbb{E}}X_{\zeta_n} + \rho^{\zeta_n}$, which implies $\mathbb{P}_n(A_n) \leq \mathbb{P}_n(X_{\zeta_n} \leq \mathbb{E}X_{\zeta_n} - \rho^{\zeta_n})$. By Chebyshev's inequality,

$$\mathbb{P}_n(A_n) \leq \mathbb{P}_n\left(\frac{X_{\zeta_n} - \mathbb{E}X_{\zeta_n}}{\sqrt{\text{Var}X_{\zeta_n}}} \leq \frac{-\rho^{\zeta_n}}{\sqrt{\text{Var}X_{\zeta_n}}}\right) \leq \frac{\text{Var}X_{\zeta_n}}{\rho^{2\zeta_n}} \to 0, \quad \text{as } n \to \infty. \quad (4)$$

By (3) and (4), we have shown that $\mathbb{P}_n$ and $\tilde{\mathbb{P}}_n$ are asymptotically orthogonal. $\qquad \square$

*Proof of Theorem 1.2.* . First of all, we have a consistent estimator $\hat{\alpha}_n$ for $\alpha$ by simply counting the number of hyperedges. If we can prove that $\beta$ can be consistently estimated by $\hat{\beta}_n = \left(2\zeta_n X_{\zeta_n} - \hat{\alpha}_n^{\zeta_n}\right)^{\frac{1}{\zeta_n}}$, then $\hat{a}_n = \frac{1}{d-1}\left(\hat{\alpha}_n + (2^{d-1}-1)\hat{\beta}_n\right)$ and $\hat{b}_n = \frac{1}{d-1}\left(\hat{\alpha}_n - \hat{\beta}_n\right)$ are consistent estimators for $a$ and $b$, respectively.

Take $\rho \in (\sqrt{\alpha}, \beta)$, by Chebyshev's inequality,

$$
\begin{aligned}
\mathbb{P}_n(2\zeta_n X_{\zeta_n} - \alpha^{\zeta_n} \geq \beta^{\zeta_n} - \rho^{\zeta_n}) &= \mathbb{P}_n\left( \frac{X_{\zeta_n} - \frac{\alpha^{\zeta_n} + \beta^{\zeta_n}}{2\zeta_n}}{\sqrt{\mathrm{Var} X_{\zeta_n}}} \geq \frac{-\frac{\rho^{\zeta_n}}{2\zeta_n}}{\sqrt{\mathrm{Var} X_{\zeta_n}}} \right) \\
&\geq 1 - \left( \frac{2\zeta_n \sqrt{\mathrm{Var} X_{\zeta_n}}}{\rho^{\zeta_n}} \right)^2 \to 1, \quad \text{as } \zeta_n \to \infty.
\end{aligned}
\tag{5}
$$

$$
\begin{aligned}
\mathbb{P}_n(2\zeta_n X_{\zeta_n} - \alpha^{\zeta_n} \leq \beta^{\zeta_n} + \rho^{\zeta_n}) &= \mathbb{P}_n\left( \frac{X_{\zeta_n} - \frac{\alpha^{\zeta_n} + \beta^{\zeta_n}}{2\zeta_n}}{\sqrt{\mathrm{Var} X_{\zeta_n}}} \leq \frac{\frac{\rho^{\zeta_n}}{2\zeta_n}}{\sqrt{\mathrm{Var} X_{\zeta_n}}} \right) \\
&\geq 1 - \left( \frac{2\zeta_n \sqrt{\mathrm{Var} X_{\zeta_n}}}{\rho^{\zeta_n}} \right)^2 \to 1, \quad \text{as } \zeta_n \to \infty.
\end{aligned}
\tag{6}
$$

Thus, $2\zeta_n X_{\zeta_n} - \alpha^{\zeta_n} \in [\beta^{\zeta_n} - \rho^{\zeta_n}, \beta^{\zeta_n} + \rho^{\zeta_n}]$ a.a.s. Since $\rho^{\zeta_n} = o(\beta^{\zeta_n})$ as $\zeta_n \to \infty$, we have $2\zeta_n X_{\zeta_n} - \alpha^{\zeta_n} = (1 + o(1))\beta^{\zeta_n}$ a.a.s. Since $\hat{\alpha}_n^{\zeta_n} \to \alpha^{\zeta_n}$ as $\zeta_n \to \infty$, we have $2\zeta_n X_{\zeta_n} - \hat{\alpha}_n^{\zeta_n} = (1 + o(1))\beta^{\zeta_n}$ a.a.s. Therefore, $\hat{\beta}_n = (2\zeta_n X_{\zeta_n} - \hat{\alpha}_n^{\zeta_n})^{\frac{1}{\zeta_n}} \to \beta$ as $\zeta_n \to \infty$. $\qquad\square$

## 3 Building blocks for proving Theorem 2.1

The main purpose of this section is to provide some building blocks for proving Theorem 2.1: Note, the techniques originate from [34]. See also Lemma 6.3 on Page 28 and some proof techniques on Page 24 in [33].

The proof that the number of loose cycles in hypergraphs follows the Poisson distribution is very different from that in the graph case. In [22], since $k$ vertices in a graph forming a cycle of length $m$ follow the Binomial distribution, we can calculate the probability $\mathbb{P}(N = m) = \mathbb{P}\left(\text{Binom}\left(k-1, \frac{1}{2}\right) \in \{m-1, m\}\right)$ for even $m$ and the probability of these vertices form a cycle is the summation of $\mathbb{P}(N = m)$ over all possible $m$ (see the proof of Lemma 1 in [22]). In the hypergraph case, the distribution of $k$ loose cycles of length $\zeta_n$ forming a cycle is no longer the Binomial distribution.

**Lemma 3.1** (Lemma 6.3 in [33]). *For any $i_1, i_2, \cdots, i_d \in \{+1, -1\}$, let $I_{i_1 i_2 \cdots i_d} := (a - b)I(i_1 = i_2 = \cdots = i_d) + b$, then for $j \geq 1$,*

$$
\begin{aligned}
F &:= \sum_{i_1, \cdots, i_{j(d-1)} \in \{+1, -1\}} I_{i_1 i_2 \cdots i_d} I_{i_d i_{d+1} \cdots i_{2d-1}} \cdots I_{i_{(j-1)d-(j-2)} i_{(j-1)d-(j-3)} \cdots i_{j(d-1)} i_1} \\
&= (a - b)^j + (a + (2^{d-1} - 1)b)^j.
\end{aligned}
$$

*Proof.* Let $J_j := (i_{(j-1)d-j+3, \cdots, i_{j(d-1)}})$, then

$$
\begin{aligned}
F &= \sum_{J_1, \cdots, J_j} \sum_{i_1, i_d, \cdots, i_{(j-1)d-(j-2)} \in \{+1, -1\}} I_{i_1 J_1 i_d} I_{i_d J_2 i_{2d-1}} \cdots I_{i_{(j-1)d-(j-2)} J_j i_1} \\
&= \sum_{J_1, \cdots, J_j} Tr(I(J_1) I(J_2) \cdots I(J_j)),
\end{aligned}
$$

where $I(J_s) = (I_{i J_s h})_{i, h = -1}^1$ is a $2 \times 2$ matrix defined as follows,

$$
\begin{aligned}
I(J_s) &= \begin{bmatrix} a & b \\ b & b \end{bmatrix} + \begin{bmatrix} b & b \\ b & a \end{bmatrix} + \sum_{J_s: \text{elements are different}} I(J_s) \\
&= \begin{bmatrix} a + b & 2b \\ 2b & a + b \end{bmatrix} + (2^{d-2} - 2) \begin{bmatrix} b & b \\ b & b \end{bmatrix} =: I_0.
\end{aligned}
$$

Then, we have $F = Tr(I_0^j)$.

Note that $I_0 = (a - b)I^2 + 2^{d-2}bJ^*$, where $I^2$ is the $2 \times 2$ identity matrix, $J^*$ is the $2 \times 2$ matrix with all entries 1. Since the eigenvalues of $J^*$ are 2 and 0, and

$$I_0 - \lambda I^2 = (a - b - \lambda)I^2 + 2^{d-2}bJ^* \implies \det(I_0 - \lambda I^2) = (2^{d-2}b)^2 \det(J^* - \frac{\lambda - (a - b)}{2^{d-2}b}I^2),$$

the eigenvalues of $I_0$ are $a - b$ and $a + (2^{d-1} - 1)b$. Therefore, we finally get

$$F = (a - b)^j + (a + (2^{d-1} - 1)b)^j.$$

$\square$

*Proof of Theorem 2.1.* (1) Note that $[X_{\zeta_n}]_r$ is the number of ordered $r$-tuples of $\zeta_n$-hyperedge loose cycles. Let $(L_{\zeta_n,1}, L_{\zeta_n,2}, \cdots, L_{\zeta_n,r})$ be the $r$-tuple of $\zeta_n$-loose cycles, $S_1$ the set of $r$-tuples where the vertices of all the loose cycles are disjoint and $S_2$ the set of $r$-tuples where at least one pair of the loose cycles is not disjoint, i.e., they share some common vertices. Then the expectation of $[X_{\zeta_n}]_r$ is given by

$$\mathbb{E}[X_{\zeta_n}]_r = \sum_{(L_{\zeta_n,i}) \in S_1} \mathbb{E}I_{\cup_{i=1}^r L_{\zeta_n,i}} + \sum_{(L_{\zeta_n,i}) \in S_2} \mathbb{E}I_{\cup_{i=1}^r L_{\zeta_n,i}}, \tag{7}$$

where $I_{\cup_{i=1}^n L_{\zeta_n,i}}$ is the indicator that $L_{\zeta_n,i}$ are $\zeta_n$-loose cycles.

Let $v_1, v_2, \cdots, v_{\zeta_n(d-1)}$ be the distinct vertices and $Y$ the indicator that $v_1, v_2, \cdots, v_{\zeta_n(d-1)}$ is a $\zeta_n$-loose cycle, then $\mathbb{E}I_{L_{\zeta_n,i}} = \binom{n}{\zeta_n(d-1)} \frac{(\zeta_n(d-1)-1)!}{2((d-2)!)^{\zeta_n}} \mathbb{E}Y$. Let $\sigma$ be a random label assignment and $\mathcal{E}(L_{\zeta_n,i})$ the hyperedge set of $L_{\zeta_n,i}$, then

$$\begin{aligned}
\mathbb{E}Y &= \mathbb{E}_\sigma \prod_{\{i_1,\cdots,i_d\} \in \mathcal{E}(L_{\zeta_n,i})} \frac{I_{i_1 i_2 \cdots i_d}(\sigma)}{\binom{n}{d-1}} \\
&= \frac{1}{2^{\zeta_n(d-1)}(\binom{n}{d-1})^{\zeta_n}} \sum_{\sigma \in \{\pm 1\}^{\zeta_n(d-1)}} \prod_{\{i_1,\cdots,i_d\} \in \mathcal{E}(L_{\zeta_n,i})} I_{i_1 i_2 \cdots i_d}(\sigma),
\end{aligned} \tag{8}$$

where $I_{i_1 i_2 \cdots i_d}(\sigma)$ is the expected degree. If $\sigma_{v_1} = \cdots = \sigma_{v_d}$, then $I_{i_1 i_2 \cdots i_d}(\sigma) = \binom{n}{d-1}p_n = a$. Otherwise, $I_{i_1 i_2 \cdots i_d}(\sigma) = \binom{n}{d-1}q_n = b$. We thus write $I_{i_1 i_2 \cdots i_d}(\sigma) = (a - b)I(\sigma_{v_1} = \cdots = \sigma_{v_d}) + b$.

By Lemma 3.1 in the Appendix, we have

$$\mathbb{E}Y = \frac{1}{(\binom{n}{d-1})^{\zeta_n}} \left[ \left( \frac{a + (2^{d-1} - 1)b}{2^{d-1}} \right)^{\zeta_n} + \left( \frac{a - b}{2^{d-1}} \right)^{\zeta_n} \right], \tag{9}$$

which implies

$$\begin{aligned}
\mathbb{E}I_{L_{\zeta_n,i}} &= \binom{n}{\zeta_n(d-1)} \frac{(\zeta_n(d-1)-1)!}{((d-2)!)^{\zeta_n}} \frac{1}{2(\binom{n}{d-1})^{\zeta_n}} \left[ \left( \frac{a + (2^{d-1} - 1)b}{2^{d-1}} \right)^{\zeta_n} + \left( \frac{a - b}{2^{d-1}} \right)^{\zeta_n} \right] \\
&\sim \frac{(d-1)^{\zeta_n}}{2n^{\zeta_n(d-1)}} \left[ \left( \frac{a + (2^{d-1} - 1)b}{2^{d-1}} \right)^{\zeta_n} + \left( \frac{a - b}{2^{d-1}} \right)^{\zeta_n} \right].
\end{aligned}$$

Note that the number of elements of $S_1$ is $|S_1| = \binom{n}{m_0} \frac{m_0!}{\zeta_n^r}$, where $m_0 = \zeta_n(d - 1)r$. When $m_0 = o(\sqrt{n})$, the first term in the right side of (7) is given by

$$\begin{aligned}
|S_1| \times \mathbb{E}I_{\cup_{i=1}^r L_{\zeta_n,i}} &= |S_1| \times \prod_{i=1}^r \mathbb{E}I_{L_{\zeta_n,i}} \\
&= \frac{n!}{(n - m_0)!} \frac{1}{\zeta_n^r} \frac{(d-1)^{\zeta_n r}}{2^r n^{\zeta_n(d-1)r}} \left[ \left( \frac{a + (2^{d-1} - 1)b}{2^{d-1}} \right)^{\zeta_n} + \left( \frac{a - b}{2^{d-1}} \right)^{\zeta_n} \right]^r \\
&= \frac{n!}{(n - m_0)! n^{m_0}} \left[ \frac{1}{2\zeta_n} (\alpha^{\zeta_n} + \beta^{\zeta_n}) \right]^r \\
&\sim \left( \frac{\alpha^{\zeta_n} + \beta^{\zeta_n}}{2\zeta_n} \right)^r,
\end{aligned}$$

where have used the fact that $\frac{n!}{(n-m_0)!n^{m_0}} \to 1$ if $m_0 = o(\sqrt{n})$.

Now it remains to show that the second term in the right side of (7) converges to 0. Since $|S_2| \leq m_0^2 n^{m_0-1}$, we have

$$\sum_{(L_{\zeta_n,i})\in S_2} \mathbb{E}I_{\cup_{i=1}^r L_{\zeta_n,i}} = |S_2| \times \mathbb{E}I_{\cup_{i=1}^r L_{\zeta_n,i}}$$

$$\leq m_0^2 n^{m_0-1} \left(\frac{a}{\binom{n}{d-1}}\right)^{|\mathcal{E}(H)|} \leq m_0^2 \frac{a^{m_0}}{n} \to 0 \quad \text{as } n \to \infty$$

when $m_0 \leq c \log_a n$ for a constant $c \in (0,1)$. Therefore, $\mathbb{E}[X_{\zeta_n}]_r \to \left(\frac{\alpha^{\zeta_n}+\beta^{\zeta_n}}{2\zeta_n}\right)^r$, which completes the proof of the first result of Theorem 2.1.

(2) To distinguish the distribution used in (1), we use $\tilde{\mathbb{E}}$ to express the expectation with respect to $\mathcal{H}_d(n, \frac{p_n+(2^{d-1}-1)q_n}{2^{d-1}})$. Similar to the proof of (1), we have

$$\tilde{\mathbb{E}}[X_{\zeta_n}]_r = \sum_{(L_{\zeta_n,i})\in S_1} \tilde{\mathbb{E}}I_{\cup_{i=1}^r L_{\zeta_n,i}} + \sum_{(L_{\zeta_n,i})\in S_2} \tilde{\mathbb{E}}I_{\cup_{i=1}^r L_{\zeta_n,i}} \tag{10}$$

and

$$\tilde{\mathbb{E}}I_{L_{\zeta_n,i}} = \binom{n}{\zeta_n(d-1)} \frac{(\zeta_n(d-1)-1)!}{2((d-2)!)^{\zeta_n}(\binom{n}{d-1})^{\zeta_n}} \left(\frac{p_n+(2^{d-1}-1)q_n}{2^{d-1}}\right)^{\zeta_n}$$

$$\sim \frac{(d-1)^{\zeta_n}}{2n^{\zeta_n(d-1)}} \left(\frac{a+(2^{d-1}-1)b}{2^{d-1}}\right)^{\zeta_n}.$$

The first term in the right side of (10) is given by

$$|S_1| \times \tilde{\mathbb{E}}I_{\cup_{i=1}^r L_{\zeta_n,i}} = \binom{n}{m_0} \frac{m_0!}{\zeta_n^r} \frac{(d-1)^{\zeta_n r}}{2^r n^{\zeta_n(d-1)r}} \left(\frac{a+(2^{d-1}-1)b}{2^{d-1}}\right)^{\zeta_n r}$$

$$= \frac{n!}{(n-m_0)!n^{m_0}} \left(\frac{\alpha^{\zeta_n}}{2\zeta_n}\right)^r$$

$$\sim \left(\frac{\alpha^{\zeta_n}}{2\zeta_n}\right)^r.$$

For $(L_{\zeta_n,i}) \in S_2$, $L := \cup_{i=1}^r$ has at most $\zeta_n(d-1)r - 1$ vertices and $\zeta_n r$ hyperedges, which implies $|\mathcal{V}(L)| < (d-1)|\mathcal{E}(L)|$. Then, we have

$$\tilde{\mathbb{E}}I_{\cup_{i=1}^r L_{\zeta_n,i}} \leq \left(\frac{a+(2^{d-1}-1)b}{\binom{n}{d-1}2^{d-1}}\right)^{|\mathcal{E}(L)|} \leq \left(\frac{a}{\binom{n}{d-1}}\right)^{|\mathcal{E}(L)|}$$

$$\leq ((d-1)!)^{|\mathcal{E}(L)|}\left(\frac{a}{n^{d-1}}\right)^{|\mathcal{E}(L)|},$$

where we have assumed $a > b > 0$. Since there are $\binom{n}{|\mathcal{V}(L)|}|\mathcal{V}(L)|!$ many hyperedges isomorphic to $L$, we get the estimation of the second term in the right side of (10) as follows,

$$\sum_{(L_{\zeta_n,i})\in S_2} \tilde{\mathbb{E}}I_{\cup_{i=1}^r L_{\zeta_n,i}} \leq ((d-1)!)^{|\mathcal{E}(L)|}\left(\frac{a}{n^{d-1}}\right)^{|\mathcal{E}(L)|}\binom{n}{|\mathcal{V}(L)|}|\mathcal{V}(L)|! \to 0 \quad \text{as } n \to \infty.$$

Therefore, $\tilde{\mathbb{E}}[X_{\zeta_n}]_r \to \left(\frac{\alpha^{\zeta_n}}{2\zeta_n}\right)^r$, which completes the proof of the second result of Theorem 2.1. $\quad\square$

## 4  Discussion on non-reconstruction of the community structures

Motivated by Theorem 1 in [22], we have the following conjecture.

**Conjecture 4.1.** *If $\beta^2 \leq \alpha$, then for any fixed vertices $v_1$ and $v_2$, $H \sim \mathcal{H}_d(n, p_n, q_n)$,*

$$\lim_{n \to \infty} \mathbb{P}_n(\tau_{v_1} = +1 | H, \tau_{v_2}) = \frac{1}{2}.$$

In the case of graph based SBM, the core strategy is to find a connection between $\mathcal{H}_2(n, p_n, q_n)$ and a Markov processes on a Galton-Watson tree. Before going further, we give the definition of a *Galton-Watson tree*. Given $\epsilon \in [0, 1)$, let $T$ be an infinite rooted tree with root $\rho$, the spin $\tau_\rho$ of the root $\rho$ is uniformly assigned from $\{+1, -1\}$. Then, conditionally independently on $\tau_\rho$, the spins $\tau_v$ of every children $v$ of $\rho$ is assigned as $\tau_v = \tau_\rho$ with probability $\epsilon$ and $\tau_v = -\tau_\rho$ with probability $1 - \epsilon$. Continuing this process will give a Galton-Watson tree with spins. The broadcasting problem on $(T, \rho, \tau)$ is whether the spin $\tau_\rho$ could be deduced from the spins at generation $l$ of the tree, where $l$ is sufficiently large. Such a problem on general infinite trees was initially proposed by Kesten and Stigum in 1966 [18], although they used somewhat different terminologies. This problem was solved by Blecher et al. [3] and Evans et al. [10].

In the case of HSBM, Pal and Zhu [25] generalized the above defined Galton-Watson tree to the so-called *multi-type Galton-Watson hypertree*. Let $T$ be an infinite rooted hypertree with root $\rho$ constructed as follows: Generate a root $\rho$ with spin $\tau_\rho \in \{+1, -1\}$ and then generate $\mathrm{Pois}(\alpha)$ many hyperedges that only intersects at $\rho$. For $0 \leq s \leq d - 1$, a hyperedge is called *type $s$* if there are $s$ many children (i.e., the vertices in the hyperedge) with spin $\tau_\rho$ and $d - 1 - s$ many children with spin $-\tau_\rho$ in the hyperedge. Hyperedges of type $d - 1$ are generated with probability $\frac{(d-1)a}{\alpha 2^{d-1}}$ and of type $s$ with probability $\frac{(d-1)b\binom{d-1}{s}}{\alpha 2^{d-1}}$ for $0 \leq s \leq d - 2$. It is easy to check that

$$\frac{(d-1)a}{\alpha 2^{d-1}} + \sum_{s=0}^{d-2} \frac{(d-1)b\binom{d-1}{s}}{\alpha 2^{d-1}} = 1.$$

We generate hyperedges of different types i.i.d. by Poisson distribution. For each hyperedge of type $s$, in the first generation, we uniformly and randomly pick $s$ vertices with spin $\tau_\rho$ and the rest of the $d - 1 - s$ vertices with spin $-\tau_\rho$. The subsequent generations are generated by induction.

Note we cannot use Theorem 1.1 in [10] directly to show that the threshold on deducing the spin of the root based the spin of vertex at long generations. A key difference between the Galton-Watson tree and the hypertree analogy is that the vertex spins in each hyperedge are not independent conditioned on the previous generations. In other words, the independence we need is the paths from the root to a vertex with the spins $+1$ and $-1$ that are independent. Although it is the case for vertices belonging to different hyperedges (by the definition above), this cannot be the case for two vertices belonging to the same hyperedge. Therefore, there is a key barrier to analyzing the broadcasting problem and relating that to the percolation model analyzed in [10]. We are very grateful to Ludovic Stephan and Yizhe Zhu for pointing out the key difference (personal correspondence).

The statement $\mathbb{P}_n(\tau_{v_1} = +1 | H, \tau_{v_2}) \to \frac{1}{2}$ is equivalent to $\mathbb{E}_n(\tau_{v_1} | H, \tau_{v_2}) \to 0$, i.e., $\mathrm{Var}_n(\tau_{v_1} | H, \tau_{v_2}) \to 1$. By the monotonicity of conditional variance, we have $\mathrm{Var}(\tau_\rho | H, \tau_v, \tau_{\partial H_l}) \leq \mathrm{Var}(\tau_\rho | H, \tau_v)$, where $\tau_{\partial H_l}$ is the spin of vertex at generation $l$ from the root $\rho$. It was shown that a neighborhood $H$ looks like a broadcasting process on a multi-type Galton-Watson hypertree (see Theorem 5.2 in [25] for more strict claim). In order to close Conjecture 4.1, in addition to overcoming the barrier above, we need to show that $\tau_v$ and $\tau_\rho$ are asymptotically almost surely conditionally independent given $\tau_{\partial H_l}$ and $H$.

## 5 Conclusion

Community detection is of great interest in many fields such as network analysis, statistics and computer vision. Hypergraphs have been widely used to model higher-order relationships among data. The SBM (graph-based) model is popular in community detection but only directly models pair-wise interactions. Its hypergraph version, HSBM, likewise serves as a benchmark for clustering algorithms on higher order interaction (hypergraph) data.

This paper provides some theoretical analysis on the community detection thresholds in the sparse HSBM. Specifically, motivated by a previous conjecture that there exists a sharp threshold on the model parameters for community detection, we proved that above the threshold, it is possible

to distinguish the hypergraph graph stochastic block model with a corresponding Erdös-Rényi hypergraph model. Importantly, we also provide consistent estimators for the model parameters, when the detection is possible.

Our estimators require the counting of the number of loose cycles in the hypergraph. To our best knowledge, there is no efficient algorithm to count the number of loose cycles in hypergraphs in general. Therefore, from the algorithmic side, it might be more practical to count the number of non-backtracking/self-avoiding walks of the same length [25, 30], which can be computed by matrix multiplication efficiently. Moreover, providing more practical estimators (or examining whether one can estimate the number of loose cycles sufficiently accurate to still use the estimators of model parameters we provide), is a worthy topic of future research.

## Acknowledgments and Disclosure of Funding

The authors would like to express their gratitude to three anonymous referees for their helpful suggestions and comments. We are very grateful for some insightful comments from Ludovic Stephan and Yizhe Zhu. This work was partly supported by the Australian Research Council grants DP200103448 (D. Suter & E. Zhang).

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
