# OpenReview forum: "Sparse Hypergraph Community Detection Thresholds in Stochastic Block Model"
_NeurIPS.cc/2022/Conference — NeurIPS 2022 Accept_

### Official Review · Reviewer_e8Wi · 2022-06-25

**Rating:** 4
**Confidence:** 3
**Soundness:** 3 good
**Presentation:** 2 fair
**Contribution:** 2 fair

**Summary:**

This paper studies the problem of community detection in hypergraph stochastic block models. The paper gives a proof of the detection threshold proposed by Angelini et al. [2].

The positive direction has previously been shown by Pal and Zhu [26], and this paper gives a new proof of the same result based on counting loose cycles in the hypergraph.

The negative direction was not previously known, and this paper generalises the proof of Mossel et al. [23] from graphs to hypergraphs.

**Questions:**

## Questions
My key question, to help judge the significance of this paper is: what are the key differences between the proof in this paper, and the one in [23]? What are the challenges in generalizing the result to hypergraphs, and how does this paper solve them?

## Minor Comments
* Line 13 - the first sentence is not grammatically correct. It could be "Community detection, or clustering, *aims* ... population. *It* is a *fundamentally* important ..."
* Lines 14 + 26 - I think there should not be colons here. I suggest just starting a new sentence.
* Line 28 typo: 'spectral' -> 'spectrum'
* Line 30 typo: 'known as *the* planted...'
* Line 37 typo: 'called *the* hypergraph...'
* Line 43: the number of vertices in a hyperedge is usually called the 'rank' of the hyperedge, and 'degree' always refers to a vertex. For example, Line 77 is ambiguous, because this paper overloads the term 'degree'.
* Lines 54-55: this is a little vague. Which computations on an adjacency tensor are NP-hard?
* Line 72 typo: "up a sign" -> "up to a sign"
* Line 97 - "see Theorem 1.1" should be in parentheses
* Figure 1 - I believe the caption is the wrong way round. Detection is impossible **above** the dashed curves, and possible **below** the dashed curves.
* Line 186 - it is not immediately clear what the 'children' of a hyperedge are. Does this refer to the vertices in the hyperedge?
* Line 189, and throughout the paper. It is quite difficult to follow inline math with many fractions and subscripts. I would recommend re-formatting to avoid inline fractions (like this $\frac{1}{2}$ ), either by rewritting as $1/2$ or using a full equation environment for more complex expressions.
* Line 209 typo: *and* also equivalent to
* Line 210 typo: 'variation' -> 'variance'
* Line 211-215: this is quite difficult to follow
* Line 216: what does it mean that "$V_2$ separates $V_1$ and $V_3$"?
* Line 222: there is a large gap after "a.a.s." - you can fix this by using "a.a.s.\ " in Latex.
* Line 231: brackets are not large enough to surround their contents. You can use "\left(" and "\right)" in latex. Though also see my earlier comment on inline math.
* Between lines 231 and 232 - the notation on this line is a bit confusing.


**Limitations:**

The authors have addressed the limitations of their work.

**Strengths And Weaknesses:**

## Strengths
* This paper studies an interesting problem which is of interest to the community.
* The presented result is new, and the proof appears to be sound.

## Weaknesses
* This paper is quite difficult to follow. It is very mathematically dense and would benefit from some additional explanation and discussion. There are also some careless errors - for example the caption of Figure 1 (see the detailed comments).
* The proof follows the proof for the graph case (Mossel et al. [23]) very closely. It is not clear how the proof differs. It would be helpful to include more discussion of the key ideas which are needed to generalise the proof to hypergraphs. This would help to determine the significance of the result.

---

> ### Author Response · Authors · 2022-08-01
> **Description of challenges**
>
> (This question is essentially the same as one by reviewer DJ3N and below we reproduce the same response)
>
> For $d=2$ (a 2-uniform hypergraph is a graph), our main results (Theorems 1.1, 1.2 and 1.3) are those that appeared in Mossel, Neeman and Sly's work (see Theorems 1, 2 and 3 in [23]). However, it is not trivial to generalise their proofs (though we may take guidance on the higher level strategy to prove, from those works). The main challenges (and how we overcame them) are summarised as follows:
>
> (1) The proof that the number of `loose cycles' in hypergraphs follows the Poisson distribution is much harder than that in the graph case. In [23], since $k$ vertices in a graph forming a cycle of length $m$ follow the Binomial distribution, we can calculate the probability $P(N=m)=P\left({\rm{Binom}}\left(k-1,\frac{1}{2}\right)\in\{m-1,m\}\right)$ for even $m$ and the probability of these vertices form a cycle is the summation of $P(N=m)$ over all possible $m$ (see the proof of Lemma 1 in [23]). In the hypergraph case, the distribution of $k$ loose cycles of length $\zeta_n$ forming a cycle is no longer the Binomial distribution and it is not easy to derive its distribution. To overcome this, we construct an indicator function and denote the condition of forming a cycle by the multiplication of some indicator functions, which can be reformulated as the trace of the multiplication of some $2\times 2$ matrices. Then, we get the multiplication result by finding the eigenvalues of the matrix (see the proof of Theorem 2.1 for details). Moreover, the distribution of cycles in Erd{\"o}s-R{\'e}nyi random graphs is a standard fact as claimed in [23], however, to the authors' best knowledge, the result in Erd{\"o}s-R{\'e}nyi random hypergraphs is not known, and needs careful analysis.
>
> (2) The verification of condition (C2) of Theorem 3.3 in the hypergraph case is very different from the graph case. In [23], the authors estimated the expectation $EY_n 1_H$ in three steps, where $1_H$ indicates any particular union of cycles occurs in the graph $G_n$. They first obtained a general formula for $EY_n 1_H$ in terms of $H$, and then applied the formula to the case where $H$ is a single cycle. Finally, they extended to the case where $H$ is a union of vertex-disjoint cycles. With the estimation of $EY_n 1_H$, they were able to compute $EY_n[X_3]_{j_3}\cdots {[X_m]}_j$ by a standard counting argument (see Chapter 4 in Bollobas's book "Random Graph"). Refer to Lemmas 11-15 in [23] for details. In the present paper, we estimate the expectation $\tilde{E}Y_n 1_L$, where $1_L$ indicates the union of $\zeta$-loose cycles $L_\zeta$, by a two-step decomposition. The first one is decomposing by whether the tuples (union of loose cycles) have a common vertex, and the second one is that by the spins of vertices. For the case where vertices have $+1$ spins and their tuples do not share a common vertex, we have used the technique that is developed in the proof of Theorem 2.1. Refer to the proof of Theorem 1.3 in the appendix for more details.
>
> (3) The most challenging proof is the verification of condition (C4) of Theorem 3.3 in this paper, which is also very different from the graph case. In [23], due to the Binomial distribution, they estimated the second moment of $Y_n$ by considering the connection between two vertices and using the technique of Taylor series (see Lemmas 7, 8, 9, 10 in [23]). In the present paper, we estimated the second moment of $Y_n$ by denoting it as an exponential function involving indicator functions of spins of vertices. The key technique is to decompose the indicator $I[\sigma_{i_1}=\sigma_{i_2}=\cdots=\sigma_{i_d}]$ as individual multiplications $\prod_j I[\sigma_{i_j}]$ and discuss different cases $\sigma_{i_j}=+1$ or $-1$. To simplify the difficulty of analysis, we only consider the leading terms up to order $n^{d-1}$ (other terms in the final expression approach to zero when we take limitation). Refer to the proof of Theorem 1.3 in the appendix for details.
>
> Last but not least, we addressed one big difficulty of transferring the HSBM problem to the hypertree problem by following Pal and Zhu's work [26]. They defined the multi-type Galton-Watson hypertree and proved the coupling between the hypergraph and hypertree. Unfortunately, there are weak long-range interactions. However, these interactions are sufficiently weak that we can achieve an asymptotic independence result (see Lemma 3.2). While we leveraged the above mentioned work for this part, it is still a contribution of this paper to apply it here.
>
> We have now revised the paper by taking the reviewer's minor comments. Apologies for some (grammar) mistakes and typos. Thank you so much for your valuable time and effort!

---

> > ### Comment · Reviewer_e8Wi · 2022-08-07
> > **Thank you for your response**
> >
> > Thank you for your detailed response regarding the challenges of generalizing the proofs from graphs to hypergraphs. I feel that the contribution of this paper is interesting, and non trivial. However, I still feel that this paper would be greatly improved with some further editing and proof reading to improve the clarity of the writing.
> >
> > For example, although the authors have updated the caption of Figure 1, the second sentence in the modified caption now doesn't seem to be true - in the figure, the lines with larger values of $d$ are lower, giving a smaller area of possible detection. It is difficult to check what is true due to the complexity of the given equation in terms of $a$ and $b$.
> >
> > My main concern here is not that there is a simple typo in the caption of the figure. My concern is that I cannot have confidence that there are not similar mistakes throughout the paper which could mislead the reader.

---

> > > ### Author Response · Authors · 2022-08-08
> > > **Thank you for your comment**
> > >
> > > Thank you for pointing out this typo and we apologize for the oversight. The correct sentence as you have pointed out should be “The larger d is, the larger the area of impossible detection is,..”. We have scanned the rest of the revised manuscript and to the best of our knowledge did not find any more typos.

---

> > > > ### Author Response · Authors · 2022-08-08
> > > > **Extra context - included  to explain to a wider readership, in particular.**
> > > >
> > > > In our initial revision we made only two of the three required changes to fix the caption: all consequential on the same inverted English expression in the original manuscript.
> > > > Specifically we should have changed “The larger d is, the larger the area of possible detection is,..” To “The larger d is, the larger the area of impossible detection is,..” in keeping with the logic that below the dashed lines are the regions of possible detection. Of course, it is embarrassing to have missed this and we sincerely apologise.
> > > >
> > > > Figure 1 was only included to help the reader obtain some intuition of the *ramifications* of our results *under varying d*. It does not form any part of the analysis. Yes, if it were included in the original form it would have confused the reader and we are very grateful to the reviewer for spotting that, and giving us a chance to correct.

---

### Official Review · Reviewer_pS3F · 2022-07-02

**Rating:** 7
**Confidence:** 4
**Soundness:** 4 excellent
**Presentation:** 3 good
**Contribution:** 4 excellent

**Summary:**

In this paper, the authors look at the problem of when it is possible to detect latent communities in a hypergraph stochastic block model. Previously the answer was known for SBM in graphs and a similar sharp threshold was conjectured for hypergraphs as well. The authors answer both the positive part and the negative part of the conjecture. The first part is answered by showing that the distribution of the number of loose cycles approach the Poisson distribution; in that case the detection problem can be resolved if certain conditions are met. On the other hand, if the conditions are not met, then by using a Galton-Watson style argument (similar to SBM), the prove the negative  part of the conjecture.

**Questions:**

1) In L75, approximate recovery is discussed. However, the problem that is solved is that of detection and not of recovery (even approximately). Here, I am a bit confused. Can you please explain?

2) Is it not possible to show the proof of Theorem 1.1 in the finite samples regime as well?

3) I believe that the writing in Section 3 can be improved significantly. It is not clear to me at all why the difficulty in solving the hypertree problem (spin of the root) translates to difficulty in solving the detection problem in hypergraph SBM? Can you please point me to the formal proof that rigorizes this argument? For now, I think the paper simply says that the neighborhood looks similar for a particular node but this is very hand-wavy.



**Limitations:**

Yes

**Strengths And Weaknesses:**

The paper is well-written and the results are very strong theoretically. The techniques are also quite interesting and I learned a lot from the paper.

---

> ### Author Response · Authors · 2022-08-01
> **Clarify detection problem, Theorem 1.1 and the difficulty transferring**
>
> Q1: Firstly, we would like to clarify the terminologies "recovery" and "detection" (see [26]). By exact recovery, we mean finding a spin vector $\hat{\sigma}$ such that
> \begin{align*}
> \lim_{n\rightarrow \infty} P(\{\mathcal{O}_n(\hat{\sigma},\sigma)=1\}\cup \{\mathcal{O}_n(\hat{\sigma},\sigma)=-1\})=1.
> \end{align*}
>
> That is, the estimated spin vector $\hat{\sigma}$ is either exactly the true spin vector $\sigma$ or exactly the negative true spin vector $-\sigma$ asymptotically. By almost exact recovery, we mean finding a spin vector $\hat{\sigma}$ such that, for any $\epsilon>0$,
> \begin{align*}
> \lim_{n\rightarrow \infty}P(\{\vert \mathcal{O}_n(\hat{\sigma},\sigma)-1\vert>\epsilon\}\cap \{\vert \mathcal{O}_n(\hat{\sigma},\sigma)+1\vert>\epsilon\})=0.
> \end{align*}
>
> That is, the empirical overlap $\mathcal{O}_n(\hat{\sigma},\sigma)$ is in $[-1,-1+\epsilon)\cup (1-\epsilon,1]$ asymptotically almost surely. This case almost exactly recovers the community, as indicated by the terminology, compared with the definition of exact recovery. By detection, we mean finding a spin vector $\hat{\sigma}$ such that, there exists a constant $c\in(0,1)$, for any $\epsilon>0$,
>
> \begin{align*}
> \lim_{n\rightarrow \infty}P(\{\vert \mathcal{O}_n(\hat{\sigma},\sigma)-c\vert>\epsilon\}\cap \{\vert \mathcal{O}_n(\hat{\sigma},\sigma)+c\vert>\epsilon\})=0.
> \end{align*}
>
> That is, the empirical overlap $\mathcal{O}_n(\hat{\sigma},\sigma)$ is in $((-c-\epsilon,-c+\epsilon)\cup (c-\epsilon,c+\epsilon))\cap [-1,1]$ asymptotically almost surely. Intuitively, this case can detect some communities but not all of them. Secondly, we would like to make clear that we have considered the detection problem, not the recovery problem in this paper. We have not used the words "approximate recovery'' any where this paper.
>
> Q2: We believe it is not possible. Reason 1: The mathematical concept "asymptotically orthogonal" is defined asymptotically, i.e., for very large $n$, not in the finite samples regime. Reason 2: To prove Theorem 1.1, we firstly proved Theorem 2.1, i.e., the distribution of the number of loose cycles. In the finite samples regime, we believe that the statistical properties of the number of loose cycles are not easy to identify.
>
> Q3: We agree with the reviewer that Section 3 could be improved, which now has been revised. The motivation of transferring the difficulty in solving the detection problem in HSBM to that in solving the hypertree problem is from Mossel, Neeman and Sly's seminal work (see [23]) on the counterpart problem in the graph case. The formal proof that rigorizes the transferring of difficulty in hypertree and HSBM is from Pal and Zhu's seminal work (see the proof of Theorem 5.2 in [26]). Apologies for the very hand-wavy claim that the neighborhood looks similar for a particular node. Now we give the following strict mathematical definitions and descriptions.
>
> Let $(H,\rho)$ be a rooted hypergraph, i.e., a hypergraph $H$ with a distinguished vertex $\rho\in \mathcal{V}(H)$. Two rooted hypergraphs $(H_1,\rho_1)$ and $(H_2,\rho_2)$ are said to be isomorphic if and only if there is a bijection $\varphi: \mathcal{V}(H_1)\rightarrow \mathcal{V}(H_2)$ such that $\varphi(\rho_1)=\rho_2$ and $e\in \mathcal{E}(H_1)$ if and only if $\varphi(e):=\{\varphi(i) \vert i\in e\}\in \mathcal{E}(H_2)$. Let $(H,\rho,\sigma)$ be a rooted hypergraph with a spin vector $\sigma$, i.e., each vertex $i\in \mathcal{V}(H)$ is given a spin $\sigma(i)\in\{+1,-1\}$. Two rooted hypergarphs $(H_1,\rho_1,\sigma_1)$ and $(H_2,\rho_2,\sigma_2)$ are said to be spin-preserving isomorphic, denoted by $(H_1,\rho_1,\sigma_1)\equiv (H_2,\rho_2,\sigma_2)$, if and only if there is an isomorphism $\varphi:(H_1,\rho_1)\rightarrow (H_2,\rho_2)$ such that $\sigma_1(j)=\sigma_2(\varphi(j))$ for any $j\in \mathcal{V}(H_1)$. Let $(H,\rho,\sigma)_l$ be the rooted hypergraph $(H,\rho,\sigma)$ truncated at generation (distance) $l$ from $\rho$ and $(T,\rho,\tau)_l$ the rooted hypertree $(T,\rho,\tau)$ truncated at distance $l$ from $\rho$. If $l=c\log(n)$ with $c\log(\alpha)<1/4$ and $c$ is a constant, then for sufficiently large $n$,
>
> (1) if $\sigma_\rho=+1$, there exists a coupling between $(H,\rho,\sigma)$ and $(T,\rho,\tau)$ such that
> \begin{align*}
> P(\{(H,\rho,\sigma)_l\equiv (T,\rho,\tau)_l\})\geq 1-n^{-1/5}.
> \end{align*}
>
> (2) if $\sigma_\rho=-1$, there exists a coupling between $(H,\rho,\sigma)$ and $(T,\rho,-\tau)$ such that
> \begin{align*}
> P(\{(H,\rho,\sigma)_l\equiv (T,\rho,-\tau)_l\})\geq 1-n^{-1/5}.
> \end{align*}
>
> Namely, we have
> \begin{align*}
> \lim_{n\rightarrow \infty}P(\{(H,\rho,\sigma)_l\equiv (T,\rho,\sigma_\rho\cdot \tau)_l\})=1,
> \end{align*}
>
> which means a neighborhood in $H$ looks like a broadcasting process on a multi-type Galton-Watson hypertree $T$.
>
> Thank you very much for your constructive suggestions and comments!

---

### Official Review · Reviewer_DJ3N · 2022-07-11

**Rating:** 6
**Confidence:** 3
**Soundness:** 4 excellent
**Presentation:** 2 fair
**Contribution:** 4 excellent

**Summary:**

This paper considers community detection in the Hypergraph Stochastic Block Model (HSBM), an extension of the popular SBM for graphs. The paper is concerned with the detection problem; namely, when is it possible to distinguish the HSBM from its Erdos-Renyi counterpart? Prior work by Pal et al. proves a result in the positive direction; when the model parameters satisfy a certain inequality, then detection is possible. The paper gives an alternate proof of the same result by relating the community detection problem to the problem of broadcasting on hypertrees (in a similar vein to the work of Mossel et al. in the SBM). This paper also gives consistent estimates of the model parameters whenever detection is possible. Finally, the paper proves a negative result, completing the conjecture of Angelini et al. (2015).

**Questions:**

•	Please comment on whether your hypertree analysis techniques apply to any broadcasting/percolation problems outside the scope of your paper.

•	It would be valuable to include a description of the technical challenges and how you overcame them.



**Limitations:**

Yes

**Strengths And Weaknesses:**

•	The proof of the positive result seems to be much simpler than the existing result of Pal et al. Even though the result is not novel, this is a valuable contribution. Moreover, the consistent estimation results are novel.

•	The negative result is significant because it confirms a conjecture, identifying the detection threshold.

•	The positive result follows by analyzing the number of “loose cycles” in the HSBM and its Erdos-Renyi counterpart, showing that the loose cycle distributions differ.

•	The negative result applies an existing result to prove contiguity of the two hypergraph distributions; the main effort is in verifying its conditions.

•	There are pervasive writing issues; please proofread carefully! The writing issues are my main concern.

•	Lemma 3.4 is hard to parse.

---

> ### Author Response · Authors · 2022-08-01
> **Comment on hypertree analysis techniques and description of challenges**
>
> Q1: Maybe the starting point is to look back at what the tree analysis techniques (or broadcasting on tree) have been applied to. In the very recent International Congress of Mathematicians, Mossel was invited to give a lecture titled "Combinatorial Statistics and the Sciences", which is publicly available on the YouTube channel of International Mathematical Union. In this talk, he mentioned two applications of the broadcasting problem on tree, one is phylogenetic inference in biology, where the phylogenetic reconstruction (infer tree structure from leaf images given a random order) is similar to the broadcast problem. The other one is the stochastic block model conjecture, whose hypergraph version problem was discussed in the present paper. Therefore, it is reasonable to suggest that our techniques/analysis might apply more widely to other applications that model data with hypergraphs.
>
> Q2: For $d=2$ (a 2-uniform hypergraph is a graph), our main results (Theorems 1.1, 1.2 and 1.3) are those that appeared in Mossel, Neeman and Sly's work (see Theorems 1, 2 and 3 in [23]). However, it is not trivial to generalise their proofs (though we may take guidance on the higher level strategy to prove, from those works). The main challenges (and how we overcame them) are summarised as follows:
>
> (1) The proof that the number of `loose cycles' in hypergraphs follows the Poisson distribution is much harder than that in the graph case. In [23], since $k$ vertices in a graph forming a cycle of length $m$ follow the Binomial distribution, we can calculate the probability $P(N=m)=P\left({\rm{Binom}}\left(k-1,\frac{1}{2}\right)\in \{m-1,m\}\right)$ and the probability of these vertices form a cycle is the summation of $P(N=m)$ over all possible $m$ (see the proof of Lemma 1 in [23]). In the hypergraph case, the distribution of $k$ loose cycles of length $\zeta_n$ forming a cycle is no longer the Binomial distribution and it is not easy to derive its distribution. To overcome this, we construct an indicator function and denote the condition of forming a cycle by the multiplication of some indicator functions, which can be reformulated as the trace of the multiplication of some $2\times 2$ matrices. Then, we get the multiplication result by finding the eigenvalues of the matrix (see the proof of Theorem 2.1 for details). Moreover, the distribution of cycles in Erd{\"o}s-R{\'e}nyi random graphs is a standard fact as claimed in [23], however, to the authors' best knowledge, the result in Erd{\"o}s-R{\'e}nyi random hypergraphs is not known, and needs careful analysis.
>
> (2) The verification of condition (C2) of Theorem 3.3 in the hypergraph case is very different from the graph case. In [23], the authors estimated the expectation $EY_n 1_H$ in three steps, where $1_H$ indicates any particular union of cycles occurs in the graph $G_n$. They first obtained a general formula for $EY_n 1_H$ in terms of $H$, and then applied the formula to the case where $H$ is a single cycle. Finally, they extended to the case where $H$ is a union of vertex-disjoint cycles. With the estimation of $EY_n 1_H$, they were able to compute $EY_n [X_3]_{j_3}\cdots$ ${[X_m]}_jm$ by a standard counting argument (see Chapter 4 in Bollobas's book "Random Graphs"). Refer to Lemmas 11-15 in [23] for details. In the present paper, we estimate the expectation $\tilde{E}Y_n 1_L$, where $1_L$ indicates the union of $\zeta$-loose cycles $\{L_\zeta\}$, by a two-step decomposition. The first one is decomposing by whether the tuples (union of loose cycles) have a common vertex, and the second one is that by the spins of vertices. For the case where vertices have $+1$ spins and their tuples do not share a common vertex, we have used the technique that is developed in the proof of Theorem 2.1. Refer to the proof of Theorem 1.3 in the appendix for more details.
>
> (3) The most challenging proof is the verification of condition (C4) of Theorem 3.3 in this paper, which is also very different from the graph case. In [23], due to the Binomial distribution, they estimated the second moment of $Y_n$ by considering the connection between two vertices and using the technique of Taylor series (see Lemmas 7-10 in [23]). In the present paper, we estimated the second moment of $Y_n$ by denoting it as an exponential function involving indicator functions of spins of vertices. The key technique is to decompose the indicator $I[\sigma_{i_1}=\sigma_{i_2}=\cdots=\sigma_{i_d}]$ as individual multiplications $\prod_j I[\sigma_{i_j}]$ and discuss different cases $\sigma_{i_j}=+1$ or $-1$. To simplify the difficulty of analysis, we only consider the leading terms up to order $n^{d-1}$ (other terms in the final expression approach to zero when we take limitation). Refer to the proof of Theorem 1.3 in the appendix for details.
>
> Thank you for thoughtful questions!

---

> > ### Comment · Reviewer_DJ3N · 2022-08-08
> > **Reply**
> >
> > Thank you for your comments! These would be valuable points to include in an updated version.

---

### Meta-Review · Area_Chair_dCX4 · 2022-08-27

**Recommendation:** Accept
**Confidence:** Certain

**Metareview:**

This paper confirm the positive part of the conjecture of Angelini et al (2015) about the existence of a sharp threshold on the model parameters for community detection for sparse hyper-graphs. Here the author’s contribution is a simpler proof than the existing one and consistent estimation of the model parameters. The authors also prove the negative part of the conjecture. One of the main negative comments of the reviewers is that the presentation can be improved by a careful proofreading and editing. The authors should incorporate the changes suggested by the reviewers for making the paper more accessible to the general audience and point out what the new contributions are in their proof, which seems to closely follow the proof in the graph setting by Mossel et al[23].


**Award:**

No

---

### Decision · Program_Chairs · 2022-09-14

Accept